

# Rates of soil organic carbon loss from rainforest to pasture conversion in a deforestation hotspot in the Amazon basin

Valentina Lara[1], Carlos A. Sierra[2], Miguel A. Peña[1], Sebastián Ramírez[1], Diego Navarrete[3], Juan F. Phillips[4], Álvaro Duque[1]

[1]Departmento de Ciencias Forestales, Universidad Nacional de Colombia, Medellín, Colombia
[2]Max-Planck-Institute for Biogeochemistry, Jena, Germany
[3]The Nature Conservancy, Bogotá, Colombia
[4]Jardín Botánico de Bogotá José Celestino Mutis, Bogotá, Colombia

*Correspondence to*: Valentina Lara (valarar@unal.edu.co)

**Abstract.** Deforestation in the Amazon basin drives critical losses of soil organic carbon stocks (*SOC*s), but the impacts of pasture conversion in tropical hotspots of deforestation remain poorly quantified in soil depths and time. Using the chronosequence approach, we sampled mature forests and pastures from 1 to 30 years old to address variations in soil bulk density (*SBD*), soil organic carbon concentration (*SOC*c) and stocks (*SOC*s). Mass spectrometry of the isotope ratio (IRMS) was used to obtain $\delta^{13}C$ values and estimate the input of forest and pasture *SOC* to the total pool. The rainforest soil (0-30 cm) stored 83.328 Mg C ha$^{-1}$, after 30 years of pasture conversion, *SOC*s declined by 21% (66.061 Mg C ha$^{-1}$), while *SBD* increased 15% (1.215 to 1.397 g cm$^{-3}$). Soil carbon turnover was depth dependent: forest-derived *SOC*s loss rate at 10-20 cm (0.112 year$^{-1}$) was twice as fast in the upper horizon (0-10 cm; 0.063 year$^{-1}$); simultaneously, pasture-derived *SOC*s gain was faster in the upper 10 cm (0.055% year$^{-1}$) reflecting the change from forests deep roots to a pastures shallower system. After 30 years of pasture conversion, forest-derived *SOC* still represented 19.6% of the total pool in the 0-10 cm soil horizon, highlighting the important effect of previous vegetation.

## 1 Introduction

The tropical rainforest of the Amazon encompasses diverse, complex, and fragile ecosystems that play a key role in regulating the global carbon cycle (Hansen et al., 2020; Qin et al., 2021; Pan et al., 2024). Amazon forests have been estimated to store around 121-126 Pg C in woody biomass and 160 Pg C in soils (Gloor et al., 2012; Malhi et al., 2006; Jobaggy and Jackson, 2000). However, the increase in tree mortality due to more frequent climatic oscillations, such as El Niño (Phillips et al., 2009; Qin et al., 2021), and the intensive conversion from forest to pastures and croplands in the region (Don et al., 2011; Hansen et al., 2013; Nunes et al., 2022), breaks the equilibrium of the carbon balance in this important ecosystem. For example, between 1990 and 2000, it has been proposed that the carbon sink in the Amazon forest decreased by about 30 % (Brienen et al., 2015), while between 2010 and 2020 it was shown that the south-eastern Amazon turned into a net carbon source (Rosan et al., 2024). However, most of the carbon balances assessed in the Amazon basin focused on the





above-ground biomass, overlooking the effect of forest removal on the carbon soil reservoir, which can even exceed the total amount of carbon stored in the above-ground counterpart (Malhi et al., 2009; Duque et al., 2025). Incorporating the effect of forest conversion to pastures on shaping changes in soil organic carbon stock (*SOC*s) is essential to improve our understanding of the overall carbon balance and its likely influence on ongoing climate change (Wang et al., 2024).


In the tropics (Asia, Africa, and South America), the conversion of primary forests to pastures has been shown to increase soil bulk density (*SBD*) by 14.0 % (±2.2) and decrease *SOC*s by 12.1 Mg C ha$^{-1}$ (±2.3) in the first 40 cm of soil after 25 years (±3) (Don et al., 2011). However, the differences in *SOC*s between forests and pastures vary according to soil depth. Some studies report higher *SOC*s in the upper 10 cm of pasture soils compared to forests, while forest soils retain higher *SOC*s in

deeper soil layers (Eclesia et al., 2012; Oliveira et al., 2022). In the Amazon basin, forest-to-pasture conversion has also shown to increase *SBD* while *SOC* decreases (Neill et al., 1997; Valladares et al., 2011; Navarrete et al., 2016; Veldkamp et al., 2020). These findings emphasize the need to assess the direct temporal effects of intensive change in land use on soil organic carbon and bulk density in intensively deforested areas of the Amazon.

Tracking changes in soil carbon stocks over time is challenging due to the high variability in *SBD*, *SOC*s, and soil organic carbon concentration (*SOC*c) (Durrer et al., 2021). To alleviate this difficulty, a robust approach to obtain accurate estimates of *SOC*c and *SOC*s is by monitoring the shifts from forest-derived to pasture-derived carbon in post-deforestation soils with the stable carbon isotope $^{13}$C. This method uses a signal in the $\delta^{13}$C values imprinted by the distinct photosynthetic pathways of forest (C$_3$) and pasture (C$_4$) plants (Balesdent et al., 1987). Forest-derived and pasture-derived *SOC* can be estimated by

analyzing soil samples with isotope ratio mass spectrometry (IRMS) to obtain forest and pasture $\delta^{13}$C values. The change in $\delta^{13}$C allows the quantification of the contribution of forest- and pasture-derived *SOC* using simple mixed models (Cerri et al., 1985). In the Amazon basin, the forest-to-pasture conversion has enriched $\delta^{13}$C (Neill et al., 1997; Navarrete et al., 2016). Several studies reported that values of $\delta^{13}$C in forest soils increase with depth, while in pasture soils it decreases with depth (Dortzbach et al., 2022; Navarrete et al., 2016; Neill et al., 1997). $\delta^{13}$C data revealed that after 20 years of forest conversion,

39-45 % of *SOC*s already come from pastures (Neill et al., 1997). This approach provides a powerful tool to assess the dynamics of *SOC* in Amazon deforestation hotspots.

In this study, our aim is to evaluate the effects of land use change from forests to pastures as a driver of changes in *SBD*, *SOC*c, *SOC*s, and absolute forest-derived and pasture-derived *SOC*s in the following 30 years after deforestation in the

Colombian Amazon. Using the chronosequence approach in primary rainforests and pastures from 1 to 30 years of age since land use conversion, and a total of 525 soil samples, we addressed the following research questions: (1) How do soil bulk density (*SBD*), soil organic carbon concentration (*SOC*c), and stocks (*SOC*s) change over time in three soil horizon depths (0-10 cm, 10-20 cm, and 20-30 cm)? (2) What is the relative contribution of forest-derived and pasture-derived soil organic carbon stocks (*SOC*s) after deforestation?



## 2 Materials and Methods

### 2.1 Study area

This study was carried out in the municipality of Cartagena de Chaira, Caquetá department, in the northwestern Amazon of Colombia, located between 0º34'0''N 74º14'0''W and 0º38'0''N 74º12'0''W (Fig. 1). This area is located in Colombia's hottest hotspot of deforestation (Cabrera et al., 2019; IDEAM, 2022). According to Global Forest Watch, this municipality has lost 165,000 ha of rainforest in the last 20 years, where the most common forest clearing method in the region is slash and burn of mature forest (Hansen et al., 2013; Silva-Olaya et al., 2022). Total annual rainfall ranges between 3246 and 3621 mm, mean annual temperature is 24.2 ºC, and elevation ranges between 170 and 240 m asl (Karger et al., 2017). The landscape matrix is primarily composed of mature terra firme forests and pastures of *Brachiaria humidicola* or *Brachiaria decumbens* (Alarcón and Tabares, 2007). Soils are dominated by highly weathered, acid and low cation exchange capacity (CEC) Haplic ferrasols (oxisols) and Haplic acrisols (ultisols) (Navarrete et al., 2016).

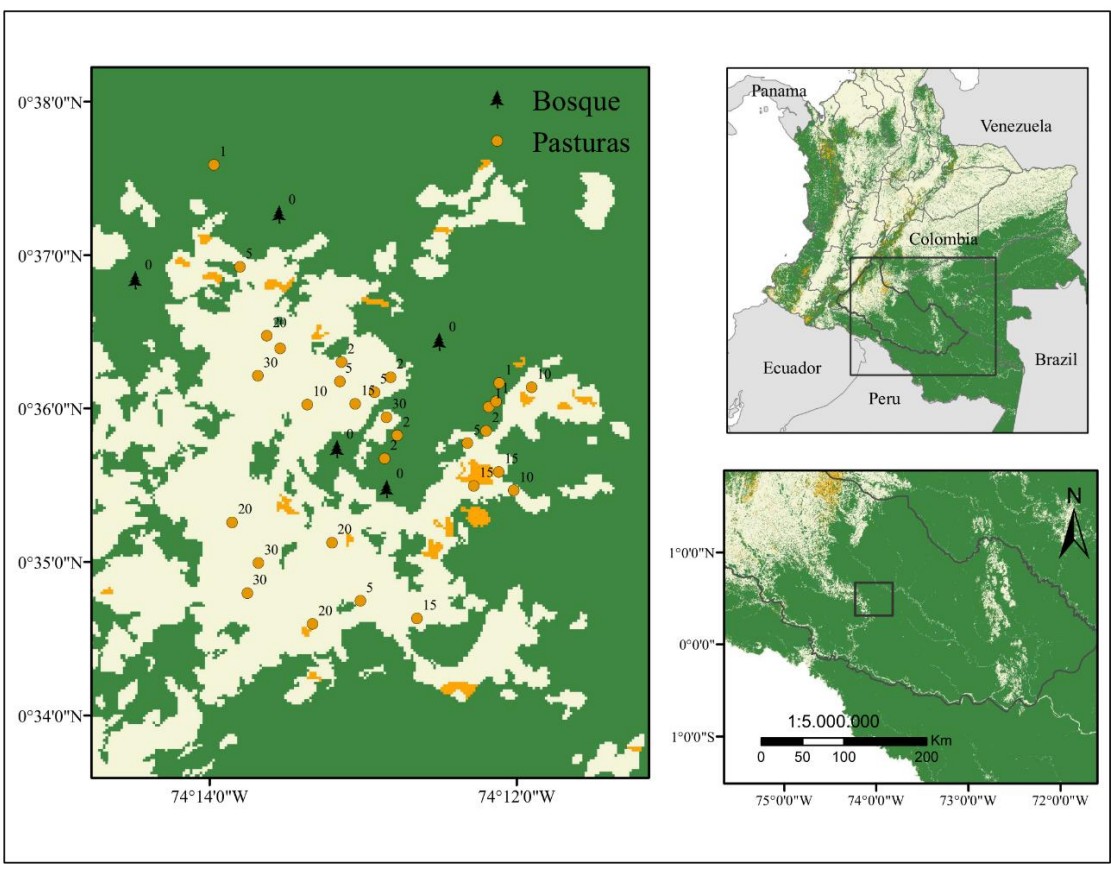

**Figure 1.** Location of conglomerates across the study area in two different land covers: forest (black icons) and pastures (yellow dots).



## 2.2 Soil sampling

The temporal evolution of carbon stock changes along a chronosequence in different types of land cover was reconstructed using data collected in pastures with ages ranging between 1 and 30 years after deforestation. Natural terra firme forests were established as the 0 year reference, while the age of the pastures was determined by personal communication with the landowners and the people who inhabit this area. In total, we established 35 conglomerates formed by five circular plots of 707 m$^2$ separated by 80 metres (Fig. 2), for a total of 175 plots. 25 plots were located in mature terra firme forest and 150 in

pastures of different ages (Table 1). Within each circular plot, a 50 x 50 cm quadrant was located 2 m away from the centre of the plot (Fig. 2). In the quadrant, the litter was removed and three soil samples were collected at depths of 0-10 cm, 10-20 cm and 20-30 cm using a 62.424 cm$^3$ cylinder, resulting in a total of 525 soil samples. The sampling was carried out in November 2017.

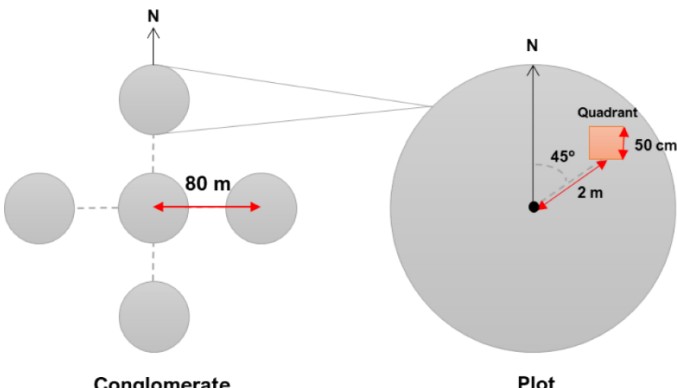


**Figure 2.** Configuration of soil sampling in conglomerate-assembled plots. Full grey circles represent plots, and the full orange rectangle shows the quadrant. Red arrows symbolize distance.

**Table 1.** Number of plots placed in terra firme forest and in pastures of different ages.

| Land cover | Age | Number of plots |
|---|---|---|
| Mature terra firme forest | 0 (reference) | 25 |
| Pasture | 1-year-old | 20 |
| Pasture | 2-year-old | 25 |
| Pasture | 5-year-old | 25 |
| Pasture | 10-year-old | 20 |
| Pasture | 15-year-old | 20 |
| Pasture | 20-year-old | 20 |
| Pasture | 30-year-old | 20 |





### 2.3 Laboratory analysis

The 525 soil samples were analyzed in the Biogeochemical Laboratory of the Universidad Nacional de Colombia sede Medellin to obtain the dry weight of the soil and quantify the soil bulk density. These samples were dried in a ventilated oven at a temperature of 55 ˚C for 12 to 24 h. Rocks and fragments of a size ≥ 5 mm in diameter and distinguishable root fragments were separated; the separated samples were placed in an oven at 105 ˚C until they reached a constant weight. Furthermore, 525 soil samples were analyzed in the Stable Isotope Facility Laboratory in the Plant Sciences Department of

the University of California – Davis, using an Elemental Vario EL Cube elemental analyzer interfaced to a PDZ Europa 20-20 isotope ratio mass spectrometer, to obtain the amount of carbon (μg) and the isotopic composition of $\delta^{13}C$ (‰). Soil samples were previously oven-dried at 65 ºC for 12 to 24 h after removing roots and rocks; then the soil was milled and stored in tin capsules.

### 2.4 Data analysis

Soil bulk density ($SBD$, g cm$^{-3}$) was quantified as $SBD = W_d/V_c$, where $W_d$ is the soil dry weight and $V_c$ is the cylinder volume. Using the C-N analyzer, the organic carbon concentration ($SOCc$, %) was calculated as $SOCc = (Ci/Wi)\times100$, where $Ci$ is the carbon content (μg) and $Wi$ is the weight of each sample (μg). Soil organic carbon stocks ($SOCs$, Mg C ha$^{-1}$) were quantified using the following Eq. (1):

$$SOC_s = SOC_c * (SBD \times 100) * d, \tag{1}$$

where $SOCc$ is the soil carbon concentration in %; $SBD$ is the soil bulk density in g cm$^{-3}$; and $d$ is the distance from the surface of the soil surface (in m) at which each sample was taken (Honorio and Baker, 2010). Total $SOC$ stocks in the topsoil (0-30 cm) were obtained as the sum of the $SOC$ stocks at 0-10 cm, 10-20 cm, and 20-30 cm (Mg C ha$^{-1}$).

A two-way analysis of variance (ANOVA) was used to evaluate the significant differences between depth (0-10 cm, 10-20

cm and 20-30 cm) and age (years after deforestation) in $SBD$ (g cm$^{-3}$), $SOCc$ (%) and $SOCs$ (Mg C ha$^{-1}$). The use of a two-way ANOVA aims to identify likely significant interactions between soil depth and age after deforestation. Tukey honestly significant tests were conducted to identify significant differences within age and depth.

We assessed the change of forest-derived ($C_f$) and pasture-derived ($C_p$) soil carbon concentration (%) over time using a

differential equation that calculates the loss of initial forest-derived carbon concentration (($C_{f0} = C_f(t = 0)$), %) after deforestation at an exponential rate -k (year$^{-1}$), which is defined as the following Eq. (2):

$$\frac{dC_f}{dt} = -k * C_{f0} \tag{2}$$



Then, we evaluated the linear accumulation of carbon concentration (%) by pastures at a constant rate r (% year$^{-1}$), as

$$\frac{dC_p}{dt} = r \qquad (3)$$

The change in total soil carbon concentration (%) is expressed as the sum of both Eq. (2) and Eq. (3). We integrated the equations and included the $\delta^{13}$C mean isotopic composition of both forests and pastures (Neill et al., 1997), obtaining the following equation Eq. (4):

$$C_t * \delta^{13}C = \left(C_{f0}{}^{-k*t} * \delta^{13}C_f\right) + ((r * t) * \delta^{13}C_p) \qquad (4)$$

This equation was fitted using a nonlinear model with the *nls* function available in the R package *stats* of R (R Core Team,
2024). Forest-derived and pasture-derived *SOC*c (%) were estimated over a 30-year chronosequence after deforestation for each soil horizon (0-10 cm, 10-20 cm and 20-30 cm). Finally, we used Eq. (1) to calculate forest-derived and pastures-derived *SOC*s in Mg C ha$^{-1}$ units.

## 3 Results

### 3.1 Changes in *SBD*, *SOC*c and *SOC*s over time and across soil horizons

The upper 30 cm of rainforest soil had on average an *SBD* of 1.215 g cm$^{-3}$ (± 0.034), *SOC*c of 2.346% (± 0.130), and *SOC*s of 83.328 Mg C ha$^{-1}$ (± 4.404). During the 30-year post-deforestation period, significant interactions between time since deforestation and soil depth were detected for *SBD* (F = 5.81; p < 0.001), *SOC*c (F = 12.21; p < 0.001) and *SOC*s (F = 9.97; p < 0.001). The 30-year-old pastures showed an increase in *SBD* at 1.397 g cm$^{-3}$ (± 0.034), along with *SOC*c and *SOC*s fell to 1.618 % (± 0.134), and 66.061 Mg C ha-1 (± 5.556) respectively. *SBD* increased rapidly during the first 2 years after forest-
to-pasture conversion and then showed a gradual increase until it reached a peak in around 20-year-old pastures. Across soil horizons, *SBD* was significantly lower in the upper 10 cm compared to the 10-20 and 20-30 cm horizons (Fig. 3a). After forest conversion to pastures, *SOC*c decreased steadily during the first two years in all soil horizons (0-10, 10-20 and 20-30 cm). The decay in *SOC*c during the first two years was then followed by an oscillatory trend that differed in magnitude across soil horizons. Average *SOC*c was significantly lower in the 20-year-old pasture compared to the rainforest (Fig. 3b).


Overall, the *SOC*s decreased by 30 % after 20 years of change in land use (0-30 cm). During the 30-year post-deforestation period, the *SOC*s exhibited two different trends. In the first 10 cm, *SOC*s decreased and increased in an unstable pattern; in contrast, on the contrary, at depths of 10-30 cm, the *SOC*s decreased steadily after deforestation. In the 10-20 cm horizon, significantly lower values of *SOC*s were found, specifically for pastures with 2, 10, 15, 20, and 30 years of age (Fig. 3c).



Most of the *SOC*s were located in the top 10 cm of soil after conversion to pastures. The 5-year-old pastures had 46 % of the *SOC*s in the upper 10 cm; this pattern continued for the older pastures, reaching 51 % in the 20-year-old pastures.

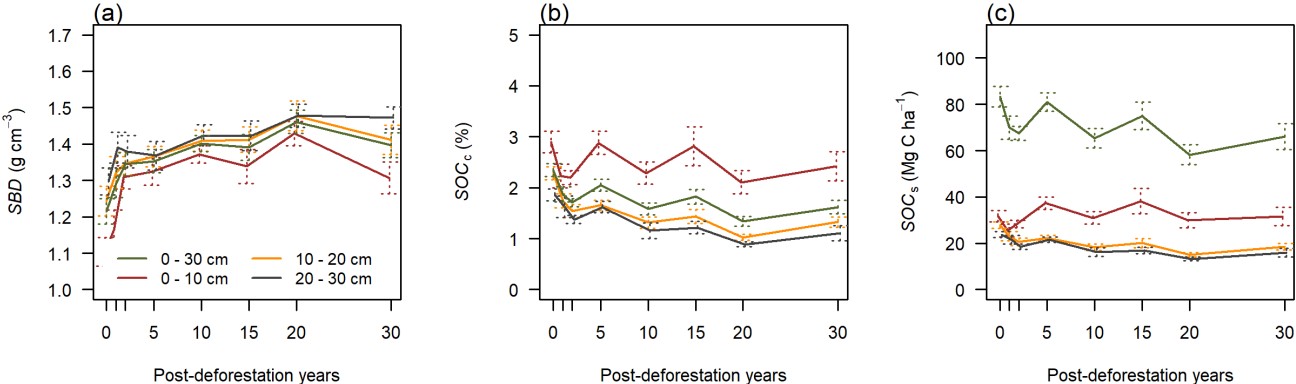

**Figure 3.** Changes in *SBD*, *SOC*c, and *SOC*s over time and across soil horizons. The lines show mean values and the dashed bars indicate the standard error. The colors represent the soil horizons: brown for 0-10 cm, orange for 10-20 cm, grey for 20-30 cm, and green for the entire depth 0-30 cm. In panels (a) and (b), the green line represents the average across the complete depth of 0-30 cm, while in panel (c), it reflects cumulative *SOC*s across all horizons.

## 3.2 Forest-derived and pasture-derived *SOC* after deforestation

The topsoil of the rainforest (0-30 cm) had a $\delta^{13}$C concentration of -28.521 ‰ (± 0.112), with a higher $\delta^{13}$C concentration in the deeper horizons (20-30 cm; -28.055 ‰). Following the change in vegetation of $C_3$ to $C_4$, $\delta^{13}$C decreased to -22.936 ‰ (± 0.361) in the 30-year-old pastures and $\delta^{13}$C was higher on the upper horizons (0-10 cm; -21.514 ‰). These significant changes in soil $\delta^{13}$C were key in estimating forest-derived and pasture-derived *SOC*. We found that forest-derived *SOC*c was lost at a higher rate (0.092 year$^{-1}$) in deeper soil horizons (20-30 cm), whereas the accumulation rate of pasture-derived *SOC*c was higher (0.055 % year$^{-1}$) in the upper 10 cm soil (Table 2). The relative contribution of forest-derived *SOC*s after 10 years of post-deforestation is greater than 60 % in all soil horizons. In the oldest pastures (30-years), forest-derived *SOC*s accounted for 19.6 % of total *SOC*s in the upper 10 cm, and 10.9 % in the deeper horizon (20-30 cm) (Fig. 4).





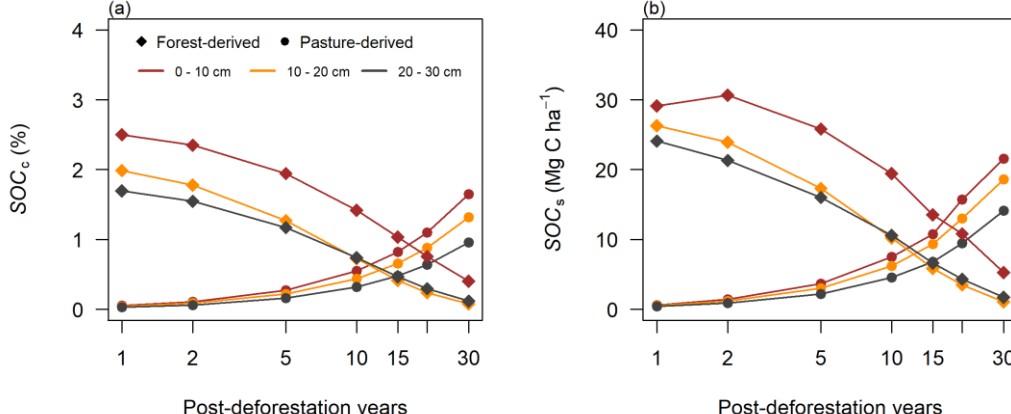

**Figure 4.** Trends of forest-derived and pasture-derived *SOC*c and *SOC*s over time and across soil horizons. Full rhombuses represent forest-derived *SOC*, and full circles represent pasture-derived *SOC*. Colors indicate soil horizons: brown for 0-10cm, orange for 10-20cm, and grey for 20-30cm.

**Table 2.** Parameter values of non-linear models fitted to the $\delta^{13}$C data to estimate forest-derived and pasture-derived soil organic carbon over time and across soil horizons.

| Soil horizon | 0-10 cm | | |
|---|---|---|---|
| Fitted model | $SOC_c * \delta^{13}\text{C} = (2.664^{-k*t} * -29) * ((r * t) * -21.5)$ | | |
| *Parameter* | *Estimate* | *Stand. Error* | *P value* |
| k | 0.063 | 0.012 | < 0.001 |
| r | 0.055 | 0.012 | < 0.001 |
| *RSE* | | 21.42 | |
| **Soil horizon** | **10-20 cm** | | |
| Fitted model | $SOC_c * \delta^{13}\text{C} = (2.223^{-k*t} * -28.6) * ((r * t) * -23.3)$ | | |
| *Parameter* | *Estimate* | *Stand. Error* | *P value* |
| k | 0.112 | 0.012 | < 0.001 |
| r | 0.044 | 0.005 | < 0.001 |
| *RSE* | | 15.18 | |
| **Soil horizon** | **20-30 cm** | | |
| Fitted model | $SOC_c * \delta^{13}\text{C} = (1.858^{-k*t} * -28) * ((r * t) * -23.9)$ | | |
| *Parameter* | *Estimate* | *Stand. Error* | *P value* |
| k | 0.092 | 0.011 | < 0.001 |
| r | 0.032 | 0.004 | < 0.001 |
| *RSE* | | 13.08 | |



## 4 Discussion

### 4.1 Changes in *SBD*, *SOC*c and *SOC*s over time and across soil horizons

Our findings demonstrate that forest-to-pasture conversion in the Colombian Amazon significantly alters soil bulk density (*SBD*) and soil organic carbon (*SOC*s) in the upper 30 cm of soil. We observed a rapid increase in *SBD* during the first years
after deforestation, followed by a gradual rise until reaching a peak around 20-year-old pastures. This pattern aligns with previous studies (e.g., Veldkamp et al. 2020), which reported a 22 % increase in *SBD* in the upper 10 cm in 25-year-old pastures. Our results also showed that *SBD* was significantly lower in the upper 10 cm compared to deeper horizons (10-20 cm and 20-30 cm), suggesting that soil compaction is higher at greater depths over time. This could be related to the reduction in organic matter input and increased mechanical pressure from grazing activities (Neill et al., 1997; Veldkamp et
al., 2020), although these factors were not directly evaluated in our study.

The observed significant decrease in *SOC*s after forest-to-pasture conversion is consistent with several studies conducted in both the eastern and south Brazilian Amazon, and across the tropics (Don et al., 2001; Damian et al., 2021; de Lima et al., 2022). However, Neill et al., (1997) found that *SOC*s increased in 14 sites located in the southwestern Amazon. This
divergence may be related to differences in soil fertility, climate, and pasture management practices (Navarrete et al., 2016). We also found that *SOC*s decline was steady in deeper soil horizons compared to the upper 10 cm, where *SOC*s exhibited an unstable pattern of decrease and increase over time. These variations between soil horizons could be explained by the alteration of microclimatic conditions and biological activity, as the heat influx in pasture soils can be up to five times higher than in forest soils (Veldkamp et al., 2020; Alvalá et al., 2002). Also, the relationship between *SBD* and *SOC*s is key to
understand this pattern, as significant degradation in *SBD* is accompanied with losses in soil C storage in the Caquetá region (Silva-Olaya *et al.*, 2022).

The variability in *SOC*s estimates can be attributed to differences in soil properties. Quesada et al. (2011) found that the western Amazon soils in Colombia and Peru, tend to have higher clay content and fertility compared to the central and
eastern Amazon. Low-fertility soils in some regions of the Amazon have shown to increase *SOC* after pasture conversion (Holmes et al., 2006; Numata et al., 2007). These findings explain why our study site exhibited an increase in *SBD* and a significant decline in *SOC*s, as clay-rich soils are more susceptible to compaction and experience faster decomposition rates under pasture conditions. Furthermore, changes in *SBD* over time can introduce uncertainty in *SOC*s estimates, as soil compression in pastures might lead to comparisons of different soil depths relative to the initial forest conditions (von Haden
et al., 2020; Fowler et al., 2023).





### 4.2 Forest-derived and pasture-derived *SOC* after deforestation

The isotopic composition of $\delta^{13}C$ in our study provided key points to understand the dynamics of forest-derived and pasture-derived *SOC*s following deforestation. In the Colombian Amazon rainforest topsoil, $\delta^{13}C$ values averaged -28.521 ‰, consistent with values reported for other Amazonian regions, such as Rondônia (-27 to -28.5 ‰; Neill et al., 1997) and the
southern Brazilian Amazon (Dortzbach et al., 2022). After forest-to-pasture conversion, we found that $\delta^{13}C$ values increased significantly, reaching an average value of -22.936 ‰ in 30-year-old pastures. This reflects the transition from $C_3$ vegetation (forest) to $C_4$ vegetation (pasture), as reported in other studies (Neill et al., 1997; Navarrete et al., 2016; Dortzbach et al., 2022).

Our results indicate that forest-derived *SOC* rate of loss in the 10-20 cm soil horizon (0.112 year$^{-1}$) was nearly twice as high compared to the upper 10 cm (0.063 year$^{-1}$). This rapid rate of forest *SOC* loss is likely due to the loss of deeper root inputs from trees following deforestation, and their replacement by pasture grasses with predominantly shallow root systems. Meanwhile, pasture-derived *SOC* accumulated more rapidly in the upper 10 cm (0.055 % year$^{-1}$) over the 30 post-deforestation years. However, the rates of forest-derived *SOC* loss vary across the Amazon basin. In Rondônia, forest-
derived *SOC*s accounted for 61 to 55 % of the total *SOC* pool after 20 years of conversion to pastures (Neill et al., 1997), whereas in our study, forest-derived *SOC*s accounted for 20% of total *SOC* stocks in the upper 10 cm after 30 years. These differences were expected due to initial carbon pool, pasture management practices, and regional climate conditions (Fujisaki et al., 2015; Trumbore et al., 1995).

The higher accumulation of pasture-derived *SOC* in the upper 10 cm can be attributed to increased humification and carbon input from $C_4$ vegetation, as well as grazing intensity. High-grazing pastures in the Caquetá region have shown greater increases in $\delta^{13}C$ compared to low-grazing pastures (Navarrete et al., 2016). The increased carbon input in the upper soil horizon can be related to changes in microbial activity and root exudates from pasture grasses (Dortzbach et al., 2022). The slower decline of forest-derived *SOC* in deeper horizons suggests that these layers are less affected by forest-to-pasture
conversion, possibly due to shallower rooting depths of pasture vegetation in comparison with the deep rooting depths of tree vegetation in the old-growth rainforest and microbial activity at greater depths.

A relevant consideration is that several studies reporting *SOC*s in pastures after deforestation often do not evaluate forest-derived and pasture-derived *SOC*s. This can lead to the misinterpretation that *SOC* remains stable or increases after forest-to-
pasture conversion, when in fact a significant portion of the total *SOC* pool may still be derived from the original mature forest (Neill et al., 1997; Fujisaki et al., 2015; Navarrete et al., 2016). Our study highlights that even after 30 years of pasture establishment, forest-derived *SOC*s still account for a substantial proportion of the total *SOC* pool. This overlooks the importance of forest-derived carbon when evaluating the long-term impacts of land-use change on *SOC* dynamics.

Overall, our results show that even though *SOC* stocks decreased by a small proportion after 30 years of deforestation, almost all the original carbon from the previous forest cover was lost and replaced by carbon of pasture origin (Fig. 3). These changes in the source of *SOC* are likely related to changes in the age and transit time of *SOC*. Previous studies have shown that carbon in the topsoil of tropical forests is on the order of decades to centuries old (Trumbore 1993, 2000, Sierra et al. 2018), while the new pasture-derived carbon is of a much more recent origin and likely more active in soils. Therefore, we

expect the soil carbon cycle to be more active in the newly established pastures with faster rates of replacement and faster return to the atmosphere in comparison to the previous forest carbon.

## 5 Acknowledgements

We acknowledge the PROYECTO GEF "Conservación de bosques y sostenibilidad en el Corazón de la Amazonia colombiana" for funding the project. Additional support was provided by Patrimonio Natural Fondo para la Biodiversidad y
Áreas Protegidas and the Universidad Nacional de Colombia.

## 6 Author contributions

VL: Data curation, Formal analysis, Writing (original draft preparation)

CAR: Supervision, Methodology, Writing (review and editing)

MAP: Supervision, Writing (review and editing)

SR: Project administration, Investigation

DN: Conceptualization

JFP: Conceptualization, Funding acquisition

AD: Conceptualization, Supervision, Writing (review and editing)

## 7 Code and data availability statement

All data and code supporting this study will be made available upon request to the corresponding author.

## 8 Conflict of Interest Statement

The authors declare no conflicts of interest.



## 9 Conclusions

This study shows that forest-to-pasture conversion in the Colombian Amazon significantly alters soil physical properties and
carbon dynamics over a 30-year period. Key findings include increased soil bulk density (*SBD*), a decline in total soil
organic carbon (*SOC*) stocks, and an accumulation of *SOC* in the upper 10 cm, alongside an increase in δ13C concentration.
Significant differences across soil horizons highlight the importance of considering soil depth when evaluating land-use
change impacts. Our results emphasize the role of pasture age, as significant variations in *SBD*, *SOC*c, and *SOC*s were found
across different pasture ages, stressing the need to account for temporal changes in land management strategies following
deforestation. These findings shed new insights for developing effective land management and restoration strategies in the
Amazon to mitigate soil degradation and promote sustainable land use, particularly in hotspots of deforestation like the
Caquetá region in the Colombian Amazon.

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
