# Peer review of "Rates of soil organic carbon loss from rainforest to pasture conversion in a deforestation hotspot in the Amazon basin"

_EGUsphere, 2025_

## Author Comment (AC1)

**Rates of soil organic carbon loss from rainforest to pasture conversion at a deforestation hotspot in the Amazon basin**

In this manuscript the authors present a relative straightforward study conducted in South East Colombia in which they use the chronosequence approach to assess changes in soil bulk density, soil C concentration, soil C stocks and origin of soil C based on 13C isotope analysis. They show that the soil C stocks declined over time, but surprisingly forest-derived soil C stocks losses were faster in the 10-20 cm depth interval than in the 0-10cm depth interval whereas the pasture derived soil C stocks increased faster in the 0-10 cm depth interval, which they explain with the change from deep roots in the forest to shallow roots in the pastures.

The strength of this study is that it was conducted in the field, based on a careful site selection with sufficient replication. It was conducted in an understudied area of the Amazon basin and as such it is important that these data become available to the scientific community. However, there are some critical points, that I think the authors should address before I can recommend publication.

We appreciate the reviewer for taking the time to read our study and for the valuable and constructive comments to improve our manuscript. Each point will be addressed below with our responses in blue.

-As is mentioned by the authors in the discussion (l. 202) soil texture can have a major influence on decomposition rates. Unfortunately, soil texture is never mentioned and may have had a strong influence on the results. The authors mention that soil were Ferralsols or Acricols (l. 75). Whereas these soils all have in common that they are highly weathered with a low CEC they may have substantial difference in soil texture, especially since Acrisols have undergone clay translocation. The best solution for this problem is if you can analyze back-up sample for soil texture (sand-silt-clay). At a minimum you should convince the reviewers that there were no significant differences in soil texture among the sampling sites.

Thank you for the important observation regarding soil texture. We mentioned Ferrasols and Acrisols in l.75 as these are the common soils found in the region of our study area. Fortunately, during the study we also collected samples for texture analysis. To assess the potential influence of texture, we retrieved our texture data from the 0-10 cm horizon for all sampling sites and conducted the following data analysis. We applied the arcsine transformation to the sand, silt and clay proportions to conduct a hierarchical cluster analysis where it determined that the optimal groups of textures were one (k=1, gap statistic) (Tibshirani et al., 2001), suggesting a consistent textural group across sampling sites. Most importantly, we evaluated whether texture (sand, silt or clay content) explained

any residual variance in the soil carbon estimates by testing the correlation between arcsine-transformed texture variables and the model residuals at the 0-10 cm soil horizon. We found non-significant relationship for sand, silt and clay content (Fig. 1), indicating that the variation in soil organic carbon stocks was mostly explained by the variables contained in the model (time and $\delta^{13}C$). These findings suggest that soil texture did not influence the reported results and will be further addressed in the discussion section.

[Figure]

Figure 1. Correlation between arcsine-transformed soil texture data and soil carbon model residuals. Dashed lines represent a linear regression between variables.

-To assess forest- and pasture-derived C normally you also need an 13C values of the 'endmembers' (**forest litter and pasture litter**). If I am not mistaken, this is not mentioned. Please give the 'pure' signatures of C3 and C4 that you used and explain how you assessed them.

In our analysis we used average values from 13C at 0 and 30 post-deforestation years (-29‰ and -21‰ respectively), by the assumption that 0-year was forest-dominated and 30-years was pasture-dominated. However, we agree with the reviewer comment, we have now revised our data and calculated 13C values from forest litter and pasture litter samples (~ -31‰ forest and -16‰ pasture). We employed these values in our model (Equation 4 – l.125) and found slightly higher rates of forest-derived SOC loss, meanwhile the reported pattern across soil horizons remained the same – higher rate of forest-derived SOC at 10-20 cm and higher rate of accumulation of pasture-derived SOC at 0-10 cm. Moreover, the updated model suggests that after 30 years following deforestation, pasture-derived SOC accounts for approximately 90% of the total SOC pool in the topsoil. These adjustments will be made in the manuscript; we thank you for highlighting this important point.

-In your assessment you assume that the forest was replaced with a 100% C4 pasture. However, in my experience, pure C4 pastures do not exist, especially in frontier areas like where you worked. So the C-input in the pasture, will be at least partly be from C3 plants and this contribution may even increase with pasture age if a pasture degrades and C3 bushes start to grow. I think this may be the main reason why you actually found slower forest-derived SOCc loss in the top 10 cm compared to the 10-20 cm depth interval: you assume that all C3 carbon in the top 10cm was forest-derived, but part of it is probably from C3 herbs or bushes in the pastures. There is no simply way to fix this, but you could do an analysis in which you make assumptions about the C3 carbon input and how this may have affected your results.

That is a good point, as pasture management is variable across the Colombian Amazon (Navarrete et al., 2016). Based on previous studies in the west of the Colombian Amazon, the pastures in our chronosequence are located where intensive cattle grazing systems are employed, as described by Navarrete et al., (2016). Therefore, we expect that presence of C3 weeds and shrubs don't influence our results.

-I noticed in Figure 1 that there are irregular yellow areas that have no pasture age. Could you explain what these areas are?

In Figure 1, there is an underlying raster map of forest/non-forest cover, the irregular yellow areas are regions with no data. These gaps are caused by cloud cover or missing satellite imagery that affect the land cover classification. We will clarify the figure caption in the manuscript.

-I did not see the original data (maybe I missed them). However, I think it is very important that these data will become available without restrictions and I encourage you to include them in an appendix or in a data repository. I know that you write that they will be made available upon request, however, it is possible that the first author cannot be reached in some years which would make the data unavailable.

Yes, we are currently checking data repositories to make our study data available. We appreciate the suggestion.

**References**

Tibshirani, R., Walther, G. and Hastie, T.: Estimating the number of data clusters via the Gap statistic, Journal of the Royal Statistical Society B, 63, 411–423, 2001.

Navarrete, D., Sitch, S., Aragão, L. E. O. C., and Pedroni, L.: Conversion from forests to pastures in the Colombian Amazon leads to contrasting soil carbon dynamics depending on land management practices, Global Change Biology, 22, 3503–3517, https://doi.org/10.1111/gcb.13266, 2016.

---

## Author Comment (AC2)

**Rates of soil organic carbon loss from rainforest to pasture conversion at a deforestation hotspot in the Amazon basin**

**Referee comment #2**

We sincerely thank the reviewer for their thorough and constructive comments to significantly improve our manuscript. Each point will be addressed below with our responses in blue.

The manuscript comprises the results of an extensive soil sampling in the Colombian Amazon, which aims to understand SOC dynamics after deforestation in a time-for-space substitution approach. The topic is timely and aligns well with the journal's scope. The dataset is robust, and the analytical methods are generally sound. Yet, the manuscript does not formulate true hypotheses nor conceptually grounded aims (l. 58ff.). Instead, the questions posed remain largely descriptive and are not connected to the processes or mechanisms that would explain the observed patterns. As a result, the study describes what is being examined but not why these patterns should occur. Without clearly articulated hypotheses, it is not possible to evaluate whether the results support or contradict the authors' intended framework. Hence, we suggest to the editor very major revision, but see high potential for improvements.

Thank you. We agree that the current form of the manuscript is largely descriptive and lacks testable hypotheses. We are committed to undertaking substantial revision to improve the manuscript. We propose to reframe the manuscript introduction, including statistical hypothesis:

- $H_0$: forest-derived carbon loss (k) and pasture-derived carbon gain (r) are equal to zero for each soil horizon (0-10 cm, 10-20 cm and 20-30 cm). The two-source isotopic mixing model outcome will be used to evaluate this hypothesis.

Moreover, we will restructure the introduction to clearly conceptualize the processes that could explain the observed patterns. For instance, the shift from deep-rooted forest to shallow-rooted pastures and the change in soil physical properties.

Overall, the dataset has strong potential to contribute to the understanding of land-use change and SOC dynamics in tropical systems. Yet, in its current form, the manuscript remains underdeveloped and does not reach the conceptual depth needed for publication. It gives the impression of being somewhat premature, and parts of the analysis read not yet sufficiently elaborated. There appears to be substantial expertise within the author team

that, if more fully integrated, could help enhance the conceptual framing and deepen the interpretation of the results.

We thank the reviewer for the suggested references. We consider they are valuable to enhance the interpretation of our results by addressing the ecosystem-level processes related to SOC dynamics. Including the differences of SOC input from forest and pastures root systems, changes in soil bulk density and chemical soil chemical properties influenced by land-use change.

The introduction and conclusion briefly mention SOC changes through time, but the temporal trajectory is not sufficiently analysed or discussed. We have some reservations about whether the sampled depths are sufficient to capture or explain effects of deforestation on SOC dynamics with depths, given that trees can root several meters deep while the sampling was limited to 30 cm, with detailed analyses conducted on 10 cm increments. Although the authors have clearly invested considerable effort in data collection and analysis, the results and discussion do not provide sufficient interest, some interpretations appear speculative and the line of argument is not yet fully convincing. The language is vague in many places. Also, a careful refinement of the title could help guide readers more effectively and add to the overall clarity of the manuscript. Overall, the study has promising elements, and we encourage the authors to undertake a substantial revision to enhance the clarity, depth, and interpretive strength of the manuscript before it should be considered for publication.

We thank the reviewer for these critical observations. We will further examine SOC changes through time by analyzing observed trajectories using similar approaches and discuss the concept of a new equilibrium of SOC stocks in the Amazon basin. Regarding the soil depth sampling, we acknowledge this limitation and we will discuss the observed patterns understanding deep soil carbon dynamics. However, several studies have found differences between the upper horizon (0-10 cm) and slightly deeper horizons (10-30 cm). Therefore, we consider it provides valuable insights as this top 30 cm layer is most rapidly affected by land-use change.

Also, we commit to replacing vague language and speculative statements with grounded analysis based on ecosystem processes. Following the substantial revision of the manuscript we will assess the title clarity.

**Specific comments**

The abstract is generally well written. Our main recommendation would be to rephrase the first sentence: as currently written, it oversimplifies the complex and sometimes

paradoxical evidence on SOC dynamics following forest loss in rainforests (Fujisaki et al., 2015), as well as the substantial body of existing research on this topic. We encourage the authors to include more on the background in which this study is framed, ensuring it reflects both the current state of knowledge in this area and the general aim of this study.

Thank you for the precise correction. We agree that our opening statement: *"Deforestation in the Amazon basin drives critical losses of soil organic carbon stocks"*, oversimplifies SOC dynamics following deforestation, considering that contrasting patterns have been observed in diverse studies as documented by Fujisaki et al., (2015). We will ensure to properly reframe it.

L. 22 "fragile": The fragility framing of the Amazon ecosystem may not fully capture its dynamics (Flores et al., 2024; Longo et al., 2018; Sakschewski et al., 2016). The system is broadly resilient, yet certain disturbances, particularly extensive forest conversion, can increase its vulnerability to critical thresholds

We agree that "*fragile*" can be a misleading for the reader. We will remove it and reframe it stating that "*The tropical rainforest of the Amazon encompasses diverse, complex and resilient ecosystems*" and further discuss the system vulnerability to critical ecological thresholds.

L. 22f. "[they] play a key role in regulating the global carbon cycle": Specify how. The link between local soil processes and global carbon dynamics should be described more clearly to avoid sounding overly general.

We will specify how by stating: *"[they] play a key role in regulating the global carbon cycle by sequestering $CO_2$ into aboveground biomass and soil carbon stocks, which, following deforestation are oxidized released to the atmosphere and lead to net carbon emissions that create positive feedback to climate change (Fujisaki et a., 2015; Veldkamp et al., 2020; Rosan et al., 2024)"*.

L. 27f. "it breaks the equilibrium of C balances in this important ecosystem": Please rephrase to more accurately reflect the processes that occur following forest loss. The current phrasing oversimplifies the issue. Don't hesitate to include more detailed information where appropriate.

We agree with the reviewer's point. We will rephrase this more accurately and elaborate on the disruption on the carbon cycling due to the loss of the forests carbon uptake and

exposure and accelerated mineralization of large soil carbon stocks, that often results in turning the system into a net carbon source.

In Material & Methods: Please specify the climate classification of the study region to provide clearer environmental context. Also, is potentially small- and large-scale agriculture an important driver for land-use change or is it only cattle ranching? Do you have information on the pastures' management or grazing intensity? Are they still being actively used?

We will address these points in the revised manuscript. Deforestation in the Caquetá region (Colombia) is driven primarily by pasture establishment for cattle ranching. While we do not have information on the specific current state of the pastures, they are located within a landscape characterized by high intensity grazing management. This was documented by co-authors in Navarrete et al., (2016).

L. 69 "the hottest hotspot of deforestation" – why is this important in the context of this study? It also is stated in the manuscript's title but we don't fully get why this information is important in the context of this study

Referring to the study site as a "hotspot of deforestation" is important because it highlights that it is an area with rapid and extensive land-use change in the Colombian Amazon where few research studies have quantified the impacts on soil carbon stocks. Providing this data is critical to inform urgent policy on deforestation and landscape management. We will elaborate on this context in the introduction and methods and will remove the phrase from the title in the revised manuscript.

The Results section should emphasize stronger key findings rather than describe data presented in figures and tables; latter should support the results...

We agree and will restructure the Results section to lead with the key findings.

L.182ff: It would be helpful to clarify if the loss of the litter layer after slash-and-burn affects the comparability of soil depth measurements, rather than active compaction in 20-30 cm?

This is a valid observation. The comparability of the upper 10 cm soil horizon between the forest and pastures is indeed affected by slash-and-burn disturbance, we will clarify this and distinguish this from the gradual compaction process resulting from cattle trampling.

L. 216ff: the argumentation that forest SOC is greater lost at 10-20 cm due to *deeper roots* being lost is quite contradicting, considering that tress can root up to several meter but this study only has data up to 30 cm. Can you explain why specifically in 10-20 cm (and not 20-30 cm) this effect is more pronounced in the context of root losses?

We thank the reviewer for this insightful point. The explanation was indeed contradicting. A more plausible interpretation is that the 10-20 cm forest soil layer was characterized by a labile carbon pool from fine roots and rhizodeposits – a key component of fast cycling soil carbon pool (Trumbore, 2009) which rapidly decompose once forest is removed. In contrast, > 20 cm soil layers contains greater proportion of mineral-associated SOC, leading to lower a decay rate (Rumpel & Kögel-Knabner, 2011; Schmidt et al., 2011).

L. 229ff: What about potential lag effects? Or generally slower processes? (Stahl et al. 2016)

We thank the reviewer for this important observation. The statement in L. 229ff "these layers [20-30 cm] are less affected" overlooks that this pattern can be influenced by slower processes that are associated with a longer decay trajectory. We will discuss this in line with Stahl et al., (2016).

Very well written conclusion. We particularly appreciate the flow, coherence, and the way the summary of findings connects to future directions in research.

**Technical corrections:**

We appreciate the reviewer for their careful reading and detailed suggestions. We will address every point in the revised manuscript.

- Throughout the manuscript one site is differently named: "mature forest", "forest", "rainforest". Please stick to one to avoid confusion.

  We will standardize to "forest" throughout the manuscript to avoid confusion.

- We would suggest avoiding vague and subjective qualifiers like "important" (e.g. l. 20, l. 27, l. 239, l. 267) providing more concise explanations of the arguments.

  We will remove subjective qualifiers and replace them with precise descriptions.

- Please be more consistent with the use of abbreviations (e.g. C, SOC, … ) throughout all the text.

  We will ensure consistent use of abbreviations in the revised manuscript.

- L. 12 "pastures from 1 to 30 years old" – Please rephrase that this is the "time since conversion from forest" or "time since pastures have been established in this specific area".

  We will rephrase to "pastures aged 1 to 30 years since conversion from forest".

- L. 14f. "The rainforest soil …." – this sentence is quite chaotic. Maybe wrong punctuation?

  We will improve punctuation in the sentence as: *"The rainforest soil (0-30 cm) stored 83.328 Mg C ha$^{-1}$). After 30 years of pasture conversion, SOC stocks had declined by 21% (66.061 Mg C ha$^{-1}$), while SBD had increased by 15% (from 1.215 to 1.397 g cm$^{-3}$)."*

- L. 18ff. We really liked this conclusion! Please provide some further perspectives on the meaning and contribution of these findings in a broader context.

  Thank you. We will discuss broader implications in the conclusion.

- L. 28ff: "For example…": Please rephrase to "Between 1990 and 2000, it was recorded that the carbon sink decreased by 30% in the Amazon forest (references), and could have even turned into a net C source in the SE Amazon between the years 2010 and 2020 (references)." – also is this value of 30% related to the Amazon *forest soil*, the Amazon *forest ecosystem*?

  We will rephrase as suggested and clarify that the 30% refers to the Amazon ecosystem, not solely soil.

- L. 30ff "However, ...": Please include something in the direction of "Carbon balances are mostly based on ecological approaches including allometric equations to estimate C storage for e.g. policy decisions (REDD+), but that they are also oversimplifying processes above- and belowground in regards to the soil-atmosphere interface (e.g. also C loss as $CO_2$ or $CH_4$)".

  We will elaborate on the limitations of allometric-based carbon estimates for policy and the oversimplification of soil-atmosphere fluxes, following the suggestion.

- L. 31ff "… , which can even exceed the total amount of C stored in the AG counterpart": Upon reviewing Malhi et al. (2009), it appears that Fig. 2 may not fully support the point being made, so another reference might strengthen this section. We also noted a slight inconsistency in discussing the role of carbon stored at 2–3 m depth while basing the analysis in this study on samples collected only to 30 cm.

  We thank the reviewer for this correction. The statement was based on values presented in Table 1. (Malhi et al., 2009) for Tapajós km 67, where soil carbon

stocks (0-3 m) exceed AG biomass. We will include other references to better support this (e.g. Quesada et al., 2011). Regarding the noted inconsistency, we will reframe the narrative. We intended to acknowledge the relevance of deep soil carbon stocks despite the focus in our study is the top 30 cm, which is the soil layer most immediately affected by deforestation.

- L. 36. Please change to "In the tropics of Asia, Africa and South America, …" – please use parentheses more conservatively. Also, the claim is quite general (all soils in all the tropics change by 14%?) and the content of this sentence is quite similar to l. 40ff. "In the Amazon basin…". Avoid these repetitions and redundancies.

  We will make the suggested changes and modify the redundant sentences.

- L. 42f. "These findings…" – again, too general

  We will replace this and provide a specific summary of the cited references.

- L. 45 "high variability" – in space or time? Or both?

  It referred to spatial variability. We will correct this and adjust the citation (Vanguelova et al., 2016)

- L. 46ff. The explanation of the methodology of isotopic signatures is redundant here – it should be added to the Material & methods part – and the second part of this paragraph (l. 53ff) reads more like a discussion.

  We will move the explanation to the Methods section. We agree that the sentence in l. 53ff reads like a discussion; we will rephrase it to focus on the isotopic signatures depth patterns to trace carbon dynamics.

- L. 75: Ferralsols, Oxisols, Acrisols and Ultisols start with a capital letter.

  We will capitalize soil type names.

- Fig1: Please translate the legend into English. What the yellow, beige or green areas in the map?

  We will correct the legend in Fig. 1. Green represents forest cover, beige represents non-forest cover and yellow represents areas with no data. These gaps are caused by cloud cover or missing satellite imagery that affect the land cover classification. We will clarify the figure caption in the manuscript.

- L. 81f "Natural terra firme forests were established as the …". Replace "were established" by "were set".

  We will change it to "were set as".

- L. 82f. "personal communication...": Is there any chance you can confirm this information via satellite data (e.g. historical imagery)?

  We agree that satellite data will be very useful to validate the time since forest to pasture conversion. We will attempt to validate it for the revised manuscript.

- In Chapter 2.3: Was inorganic carbon present or not? If not, please add that total C equals organic C, if yes, did you remove it before or how did you handle inorganic C?

  Inorganic carbon was not present in the soils of our study. This is consistent with the pedology of the Amazon basin, which is characterized by acidic and highly weathered soils (Quesada et al., 2011). Therefore, total C in our study refers to organic carbon. We will add this clarification to the manuscript.

- L. 95: change "sede" to "campus" – if you agree with the translation.

  We agree with the translation and will change it accordingly.

- L. 101ff. "Soil samples…": This steps' goal is unclear to us.

  The mentioned steps' goal was to prepare the sample for subsequent isotopic analysis. We will clarify this.

- L. 110f. "… d is the distance from the surface of the soil surface (in m)" please change to "d is is the soil thickness [m]"?

  We will change it to soil thickness as suggested.

- In Chapter 2.4: If applicable, please add references of equations 1 to 4.

  We will add the reference for the equations. For Eq.1 we will cite Honorio and Baker (2010) and IPCC (2006) guidelines. For Eq. 4 we will cite Balesdent et al. (1987) and Neill et al. (1997). Eq. 2 and 3 represent the standard decay and accumulation models, we will check for references.

- L. 114ff: Did you meet all assumptions of normality and homogeneity of variance for the ANOVA with your dataset? If yes, state so in text.

  We thank the reviewer for this methodological observation. We conducted analysis to test normality and homogeneity of variance for our ANOVA models. For SBD, the assumptions were met (Shapiro-Wilk and Levene's test). However, soil organic carbon concentration (SOCc) and stocks (SOCs) did not meet the assumptions. To assess this issue, we used the non-parametric Kruskal-Wallis test followed by Dunn's post-hoc test and found significant differences across time and depth. We will update the data analysis and results sections accordingly.

- L. 136: delete "during the 30-year post-deforestation period".

- L. 138: change "SBD at" to "SBD to" & "fell" to "decreased".

- 139: add a comma before "respectively".

We will conduct the suggested changes for L. 136, L. 138 and 139.

- Fig 3: Make error bars as solid lines and slightly dodged, and lines between the years dashed. Or do you assume a linear trend between the years? Improve image quality to 600 dpi, in case it is not specified otherwise by the journal.

We agree with the suggestions. We do not assume a linear trend between the years.

- L. 162: change "was higher on" to "was lower in". **Please check results with Tab.2 – they mismatch.**

We agree with the change and thank the reviewer for highlighting this confusion. The apparent mismatch arises because L. 162 described the observed average $\delta^{13}C$ values in the topsoil (0-30 cm) and each depth, while Table 2. – fitter model row presented the endmember values $\delta^{13}C_f$ and $\delta^{13}C_p$ that were established as the average value at 0 and 30 years after forest to pasture conversion at each soil horizon. Following a comment from Referee #1 we have acknowledged a methodological issue in our initial endmember signature values, $\delta^{13}C_f$ and $\delta^{13}C_p$ for the modelling. We have since revised our data analysis and quantified the endmember from forest litter and pasture litter samples ($\delta^{13}C_f \sim$ -31‰ forest and $\delta^{13}C_p \sim$ -16‰ pasture). To resolve this, we will:

  (1) Improve the clarity of L.162
  (2) Provide a table as supplementary material with $\delta^{13}C$ over time since forest conversion to pastures.
  (3) Update Table 2 with corrected endmembers and results of k and r.
  (4) Add clear explanation in the Material & Methods section.

- L. 163: change "were key in estimating" to "allowed the estimation of"

We will change it to "allowed the estimation of".

- L. 163ff: **Tab 2 shows that actually is 10-20cm the depth with the greatest forest-derived C loss and not 20-30 cm (!) - please check this!**

We thank the reviewer for the observation, the statement in L. 163ff is inaccurate. We will correct this in line with the updated Table 2.

- Table 2: Equations of fitted models are not equivalent to Equation 4. Please **check the operators between summands** - also if models were then calculated correctly.

Also, for better general understanding, please add what k and r describe in the tables' description and metadata for reproducibility in e.g. the supplements.

We agree with the observations. This was a formatting error in the table compilation. We will (1) check the code to ensure accurate simulations, (2) elaborate a clearer table with descriptions of k and r, and (3) provide a reproducible model code in the supplementary material.

**References**

Flores, B. M., Montoya, E., Sakschewski, B., Nascimento, N., Staal, A., Betts, R. A., Levis, C., Lapola, D. M., Esquível-Muelbert, A., Jakovac, C., Nobre, C. A., Oliveira, R. S., Borma, L. S., Nian, D., Boers, N., Hecht, S. B., ter Steege, H., Arieira, J., Lucas, I. L., … Hirota, M. (2024). Critical transitions in the Amazon forest system. Nature, 626(7999), 555–564. https://doi.org/10.1038/s41586-023-06970-0

Fujisaki, K., Perrin, A.-S., Desjardins, T., Bernoux, M., Balbino, L. C., & Brossard, M. (2015). From forest to cropland and pasture systems: A critical review of soil organic carbon stocks changes in Amazonia. Global Change Biology, 21(7), 2773–2786. https://doi.org/10.1111/gcb.12906

Longo, M., Knox, R. G., Levine, N. M., Alves, L. F., Bonal, D., Camargo, P. B., Fitzjarrald, D. R., Hayek, M. N., Restrepo-Coupe, N., Saleska, S. R., da Silva, R., Stark, S. C., Tapajós, R. P., Wiedemann, K. T., Zhang, K., Wofsy, S. C., & Moorcroft, P. R. (2018). Ecosystem heterogeneity and diversity mitigate Amazon forest resilience to frequent extreme droughts. New Phytologist, 219(3), 914–931. https://doi.org/10.1111/nph.15185

Sakschewski, B., von Bloh, W., Boit, A., Poorter, L., Peña-Claros, M., Heinke, J., Joshi, J., & Thonicke, K. (2016). Resilience of Amazon forests emerges from plant trait diversity. Nature Climate Change, 6(11), 1032–1036. https://doi.org/10.1038/nclimate3109

Stahl, C., Freycon, V., Fontaine, S., Dezécache, C., Ponchant, L., Picon-Cochard, C., Klumpp, K., Soussana, J.-F., & Blanfort, V. (2016). Soil carbon stocks after conversion of Amazonian tropical forest to grazed pasture: Importance of deep soil layers. Regional Environmental Change, 16(7), 2059–2069. https://doi.org/10.1007/s10113-016-0936-0

Balesdent, J., Mariotti, A., and Guillet, B.: Natural 13C abundance as a tracer for studies of soil organic matter dynamics, Soil Biology and Biochemistry, 19, 25–30, https://doi.org/10.1016/0038-0717(87)90120-9, 1987.

Trumbore, S.: Radiocarbon and Soil Carbon Dynamics, Annu. Rev. Earth Planet. Sci., 37, 47–66, https://doi.org/10.1146/annurev.earth.36.031207.124300, 2009.

Rumpel, C. and Kögel-Knabner, I.: Deep soil organic matter—a key but poorly understood component of terrestrial C cycle, Plant Soil, 338, 143–158, https://doi.org/10.1007/s11104-010-0391-5, 2011.

Schmidt, M. W. I., Torn, M. S., Abiven, S., Dittmar, T., Guggenberger, G., Janssens, I. A., Kleber, M., Kögel-Knabner, I., Lehmann, J., Manning, D. A. C., Nannipieri, P., Rasse, D. P., Weiner, S., and Trumbore, S. E.: Persistence of soil organic matter as an ecosystem property, Nature, 478, 49–56, https://doi.org/10.1038/nature10386, 2011.

IPCC. IPCC Guidelines for National Greenhouse Gas Inventories. Prepared by the National Greenhouse Gas Inventories Programme, Eggleston H.S., Buendia L., Miwa K., Ngara T., and Tanabe K. (eds), 2006.